# Effect of Non-Covalent Interactions on the 2,4- and 3,5-Dinitrobenzoate Eu-Cd Complex Structures

Maxim A. Shmelev [ID], Aleksandr S. Chistyakov, Galina A. Razgonyaeva, Vladimir V. Kovalev, Julia K. Voronina [ID], Fedor M. Dolgushin, Natalia V. Gogoleva *, Mikhail A. Kiskin [ID], Alexey A. Sidorov and Igor L. Eremenko *

N. S. Kurnakov Institute of General and Inorganic Chemistry, Russian Academy of Sciences, 31 Leninsky Prosp., 119991 Moscow, Russia; shmelevma@yandex.ru (M.A.S.); aleksandr.s.chistyakov@gmail.com (A.S.C.); razgonyaeva@igic.ras.ru (G.A.R.); kovalevlad@igic.ras.ru (V.V.K.); juliavoronina@mail.ru (J.K.V.); fedya@xray.ineos.ac.ru (F.M.D.); m_kiskin@mail.ru (M.A.K.); sidorrov@mail.ru (A.A.S.)

* Correspondence: gogolevanv@inbox.ru (N.V.G.); ilerem@igic.ras.ru (I.L.E.)

**Abstract:** Heterometallic $\{Eu_2Cd_2\}$ complexes $[Eu_2(NO_3)_2Cd_2(Phen)_2(2,4-Nbz)_8]_n \cdot 2n$MeCN (I) and $[Eu_2(MeCN)_2Cd_2(Phen)_2(3,5-Nbz)_{10}]$ (II) with the 2,4-dinitrobenzoate (2,4-Nbz) and 3,5-dinitrobenzoate (3,5-Nbz) anions and 1,10-phenanthroline were synthesized. The compounds obtained were characterized by X-ray single-crystal analysis, powder X-ray diffraction analysis, IR spectroscopy, and elemental analysis. Moreover, the thermal stability of the complexes was also studied. Analysis of the crystal packing showed that where 1,10-phenanthroline is combined with various isomers of dinitrobenzoate anions, different arrangements of non-covalent interactions are observed in the complex structures. In the case of the compound with the 2,4-dinitrobenzoate anion, these interactions lead to a significant distortion of the metal core geometry and formation of a polymeric structure, while the complex with the 3,5-dinitrobenzoate anion has a structure that is typical of similar systems. The absence of europium metal-centered luminescence at 270 nm wavelength was shown. For all the reported compounds, a thermal stability study was carried out that showed that the compounds decomposed with a significant thermal effect.

**Keywords:** cadmium-lanthanide(III) complexes; 2,4-dinitrobenzoic acid; 3,5-dinitrobenzoic acid; non-covalent interactions; coordination polymer; single crystal X-ray; thermogravimetric analysis

## 1. Introduction

The development of directed synthesis methods is a requirement for efficient search for new coordination compounds with desired physicochemical properties that would be promising for solving various practical problems. For example, it can be used to increase the sorption capacity of MOFs through postsynthetic modification [1], in design of charge-transfer complexes [2], or for varying the solubility, stability, hygroscopicity, and activity of biologically active compounds [3,4].

Step-by-step modifications of the molecule composition or geometry and crystal packing make it possible to determine the effect of various factors on the physicochemical properties of compounds and identify "structure-properties" correlations [5–9].

Our latest studies on Cd and $\{Ln^{3+}Cd\}$ pentafluorobenzoate complexes with various monodentate and chelate ligands have shown that non-covalent interactions between aromatic fragments can significantly affect the geometry of metal centers and lead to the formation of previously unknown structures. The formation of polymeric structures accompanied by a sharp change in the geometry of the metal center compared to molecular analogues leads to an almost twofold increase of the luminescent quantum yield in comparison to molecular complexes with the same composition [9,10].

When studying fluorobenzoate complexes, it is impossible to trace the influence of substituent positions in the aromatic ring, since the structure-forming effects of non-covalent interactions characteristic of pentafluorobenzoate systems do not appear when

passing to benzoic acid substituents with a fewer of fluorine atoms. In this case, compounds characteristic of benzoate systems are formed [11].

In order to determine the effect of the position of a substituent in the benzoate anion on the structure and arrangement of non-covalent interactions in the crystal packing of complexes, our attention was drawn to the 2,4-dinitrobenzoate (2,4-Nbz) and 3,5-dinitrobenzoate (3,5-Nbz) anions. Non-covalent interactions have a significant effect on the structure and crystal packing of dinitrobenzoate complexes with aromatic N-donor ligands ($\pi$-$\pi$, NO-$\pi$, $NO_2$-NO, and others) [12–15]. These interactions are characterized by a high degree of directionality, and their energy is comparable to the hydrogen bond energy. Some of these compounds may have a characteristic intense color due to charge transfer in the system [16].

Interest to {EuCd} complexes is due to the bright metal-centered luminescence in the red region of the spectrum, characteristic of this class of compounds, which can be used in biomedical imaging, in creation of photosensitive labels, sensors, OLED elements and other devices [17–22]. Although dinitrobenzoate complexes show rather little promise in the preparation of photoluminescent materials, they represent an excellent model system for studying the position of strong electron-withdrawing substituents on structure-forming effects [23–25], as well as for the structure-property correlations analysis.

This work was aimed at the preparation of analogues of {LnCd} fluorobenzoate complexes that we studied previously in order to determine the effect of non-covalent interactions on the structure of {LnCd} complexes with 2,4- and 3,5-dinitrobenzoic acid and 1,10-phenanthroline, as well as at studying the structure and thermal stability of the complexes obtained.

## 2. Experimental

### 2.1. Materials and Methods

All operations related to the synthesis of new complexes are performed in air using ethanol (96%) and acetonitrile (99.9%). Following reagents are used to obtain new compounds: $Cd(NO_3)_2 \cdot 4H_2O$ (99+%, Acros organics), $Eu(NO_3)_3 \cdot 6H_2O$ (99.99%, Lanhit), KOH (99+%, Reachim), 3,5-dinitrobenzoic acid (H(3,5-Nbz), 99.9%, Aldrich), 2,4-dinitrobenzoic acid (H(2,4-Nbz), 98%, Aldrich), phenanthroline monohydrate (Phen·$H_2O$, 99%, Alfa Aesar).

Single crystal X-Ray diffraction experiments with I and II were done on a Bruker D8 Venture diffractometer (Bruker, Billerica, MA, USA) with a CCD camera and a graphite monochromated MoK$\alpha$ radiation source ($\lambda$ = 0.71073 Å). Semiempirical absorption corrections were applied for all the experiments using SADABS [26]. Direct methods were used in structure solving. The refinement was done by the full-matrix least squares technique in anisotropic approximation for all non-hydrogen atoms. The H atoms were calculated geometrically and refined in the riding model. The calculations were performed in SHELX [27] using Olex2 [28]. The SHAPE 2.1 software [29] was used to determine the metals polyhedrons geometry. The most important experimental crystallographic data and refinement statistics for I and II are reported in Table S1. Supplementary crystallographic data for the compounds synthesized are given in CCDC numbers 2103445 (for I) and 2103442 (for II). These data can be obtained free of charge from The Cambridge Crystallographic Data Centre via www.ccdc.cam.ac.uk/data_request/cif (accessed on 2 February 2022).

The PXRD data in the form of powder patterns were collected on a Bruker D8 Advance diffractometer (Bruker, Billerica, MA, USA) with a LynxEye detector in Bragg-Brentano geometry, with the sample thinly dispersed on a zero-background Si sample holder, $\lambda$(CuK$\alpha$) = 1.54060 Å, $\theta/\theta$ scan with variable slits (irradiated length 20 mm), $2\theta$ from 5° to 35°, step size 0.02°. The obtained patterns were Rietveld refined using TOPAS 4 software.

IR spectra of the compounds were recorded on a Perkin Elmer Spectrum 65 spectrophotometer equipped with a Quest ATR Accessory (Specac) by the attenuated total reflectance (ATR) in the range 4000–400 $cm^{-1}$.

Elemental analysis of the synthesized compounds was carried out on a EuroEA 3000 (EuroVector) C,H,N,S-analyzer.

Differential scanning calorimetry (DSC) was performed on differential scanning calorimeter, DSC-60 Plus, Shimadzu. Differential thermal analysis (DTA) and thermogravimetry (TGA) were performed on synchronal thermal analyzer DTG-60, Shimadzu. All experiments were carried out in air at a heating rate of 5 °C/min.

### 2.2. Synthesis of Compounds

#### 2.2.1. [Eu$_2$(NO$_3$)$_2$Cd$_2$(Phen)$_2$(2,4-Nbz)$_8$]$_n$·2$n$MeCN (I)

Cd(NO$_3$)$_2$ · 4H$_2$O (0.100 g, 0.324 mmol) was dissolved in 15 mL EtOH, and a solution of K(2,4-Nbz) (0.162 g, 0.649 mmol) in 10 mL EtOH (prepared by neutralization of H(2,4-Nbz) (0.138 g, 0.649 mmol) with KOH (0.036 g, 0.649 mmol)) was added. The reaction mixture was stirred for 15 min at 70 °C and the white precipitate of KNO$_3$ was filtered off. A solution of Eu(NO$_3$)$_3$ · 6H$_2$O (0.049 g, 0.108 mmol) in 30 mL MeCN was added to the reaction mixture, which was then stirred for 20 min at 70 °C. Phen·H$_2$O (0.064 g, 0.324 mmol) was added to the reaction media. The resulting colorless solution was stirred for 10 min at 70 °C and allowed to evaporate at room temperature. The colorless crystals suitable for X-ray diffraction studies that precipitated after 8 days were filtered off, washed with cold MeCN ($T \approx 7$ °C) and dried in air at 20 °C. The yield was 0.049 g (33.1% based on Eu(NO$_3$)$_3$ · 6H$_2$O. Calculated for for C$_{84}$H$_{46}$N$_{24}$O$_{54}$Cd$_2$Eu$_2$: C 36.24; H 1.66; N 12.07. Found: C 36.41; H 1.41; N 12.15. IR-spectrum (ATR), $\nu$ (cm$^{-1}$): 3098 m, 2499 w, 2258 w, 2091 w, 1612 m, 1594 s, 1539 s, 1466 s, 1403 s, 1345 s, 1276 s, 1146 m, 1068 m, 1020 m, 919 w, 850 m, 786 m, 724 s, 691 w, 640 m, 528 w, and 417 m.

#### 2.2.2. [Eu$_2$(MeCN)$_2$Cd$_2$(Phen)$_2$(3,5-Nbz)$_{10}$] (II)

The synthesis of II was performed in accordance to the procedure similar to that for I, using H(3,5-Nbz) instead of H(2,4-Nbz). Colorless crystals suitable for X-ray diffraction studies were found after 10 days, filtered off, washed by cold MeCN ($T \approx 7$ °C) and dried in air at 20 °C. The yield was 0.051 g (30.9% based on Eu(NO$_3$)$_3$ · 6H$_2$O). Calculated for C$_{98}$H$_{52}$N$_{26}$O$_{60}$Cd$_2$Eu$_2$: C 38.19; H 1.70; N 11.82. Found: C 38.11; H 1.85; N 11.76. IR-spectrum (ATR), $\nu$ (cm$^{-1}$): 3095 m, 2874 w, 2278 w, 1616 s, 1580 m, 1536 s, 1513 m, 1461 m, 1341 s, 1140 w, 1110 w, 1077 m, 920 m, 847 m, 788 m, 718 s, 663 w, 528 w, 448 m, and 410 w.

## 3. Results

### 3.1. Synthesis and Structure of Complexes

Compounds [Eu$_2$(NO$_3$)$_2$Cd$_2$(Phen)$_2$(2,4-Nbz)$_8$]$_n$·2$n$MeCN (I) and [Eu$_2$(MeCN)$_2$Cd$_2$ (Phen)$_2$(3,5-Nbz)$_{10}$] (II) were obtained by the exchange reaction of Cd(NO$_3$)$_2$ · 4H$_2$O, Eu(NO$_3$)$_3$ · 6H$_2$O, and the corresponding potassium monocarboxylate in EtOH and MeCN at 70 °C followed by addition of phenanthroline monohydrate. The resulting samples were characterized by single crystal X-ray analysis, powder X-ray diffraction analysis, elemental analysis, IR spectroscopy. Moreover, the thermal stability of the complexes was studied. All the new complexes were isolated as single crystals and polycrystals. The phase purity of polycrystalline samples was confirmed by X-ray powder diffraction (Figures S1 and S2).

Complex I crystallizes with acetonitrile solvates as single crystals. The 1D polymer chain of compound I consist of {Eu$_2$Cd$_2$} tetranuclear fragments with a zigzag arrangement of metal atoms, in which the central lanthanide ions are bound to each other and to peripherical cadmium ions by $\mu_2\kappa^1\kappa^1$-, $\mu_3\kappa^1\kappa^1\kappa^1$-, and $\mu_3\kappa^1\kappa^2\kappa^1$-anions (Figure 1a). The polymer chain translation occurs parallel to crystallographic axis a, while adjacent fragments {Eu$_2$(NO$_3$)$_2$Cd$_2$(Phen)$_2$(2,4-Nbz)$_8$} are bound to each other by two $\mu_3\kappa^1\kappa^1\kappa^1$-2,4-Nbz anions. The coordination environment of europium in compound I best fits a "muffin"-type polyhedron (Table S2, Figure 2a, EuO$_9$) consisting of four bridged oxygen atoms, three chelate-bridged 2,4-NO$_2$ anions, and one chelated nitrate anion. The cadmium coordination polyhedron in compound I is an octahedron (Table S2, Figure 2a, CdO$_4$N$_2$) formed by three

chelate-bridged and one bridging carboxylate anions, and completed by two nitrogen atoms of 1,10-phenanthroline.

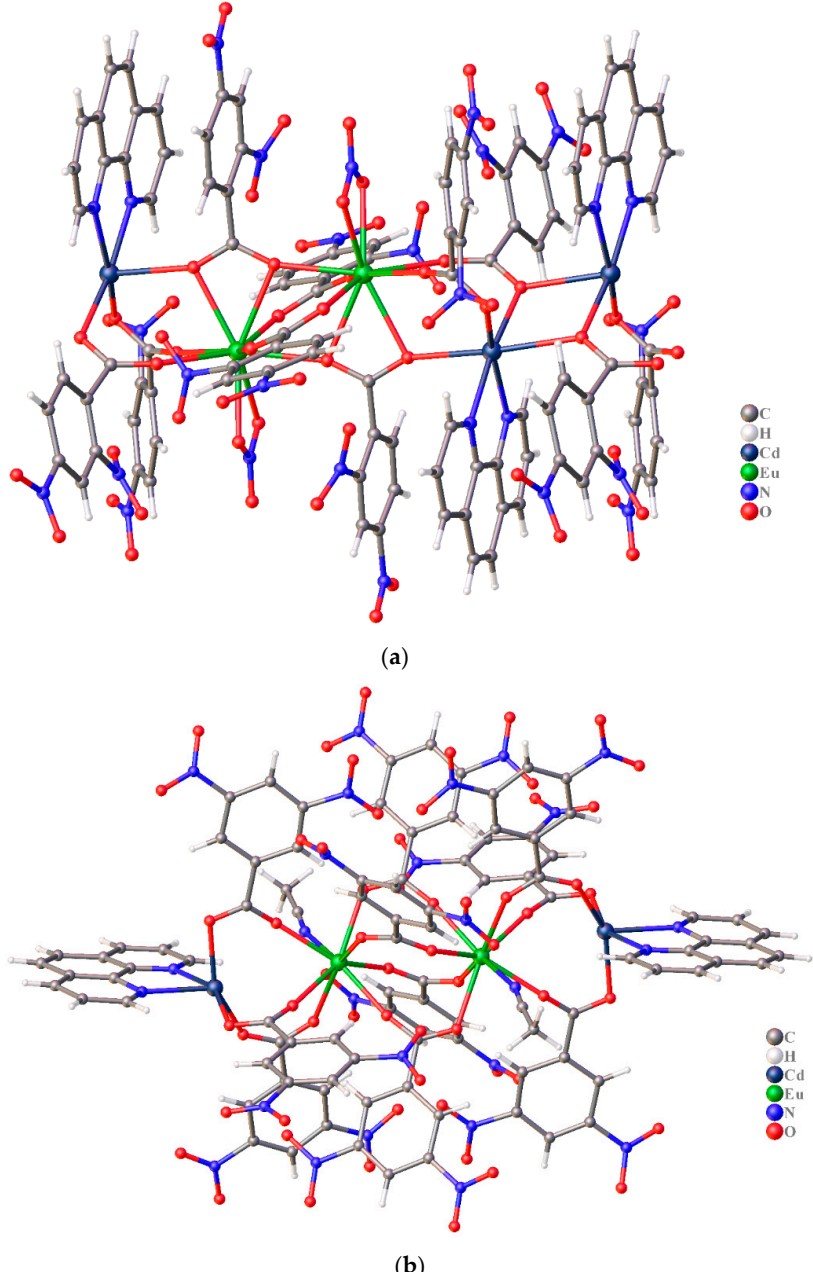

(**a**)

(**b**)

**Figure 1.** A fragment of polymeric chain of I (**a**) and structure of complex II (**b**). MeCN molecules in I are not shown.

In molecular complex II, the central europium ions are bound to each other and to terminal cadmium ions by $\mu_2\kappa^1\kappa^1$- and $\mu_3\kappa^1\kappa^1\kappa^1$-3,5-Nbz anions to form a tetranuclear linear Cd-Eu-Eu-Cd metal fragment where the metal atoms lie in the same plane (Figure 1b). The cadmium coordination polyhedron in compound II best fits a capped trigonal prism (Figure 2b, Table S2, $CdO_4N_2$) formed by two chelate-bridged and one bridging 3,5-$NO_2$ anions and completed by two nitrogen atoms of 1,10-phenanthroline (Figure 1b). The europium coordination environment in compound II best fits a square antiprism (Figure 2b), Table S2, $EuO_7N$) formed by oxygen atoms from seven bridging 3,5-$NO_2$ anions and completed by a nitrogen atom of the acetonitrile molecule. In the literature there are examples of "small" solvent molecules bound by lanthanide atoms in similar heterometallic

carboxylate complexes [30–34], however, in most compounds, rare-earth ions complete their environments by O-donor ligands.

In compound II, the polyhedra of adjacent metal ions have a common apex (Figure 2b), while the packing of polyhedra in compound I is much denser and all adjacent polyhedra share a common edge (Figure 2a). The M-O and M-N bond lengths in compounds I and II have typical values and are listed in Table 1.

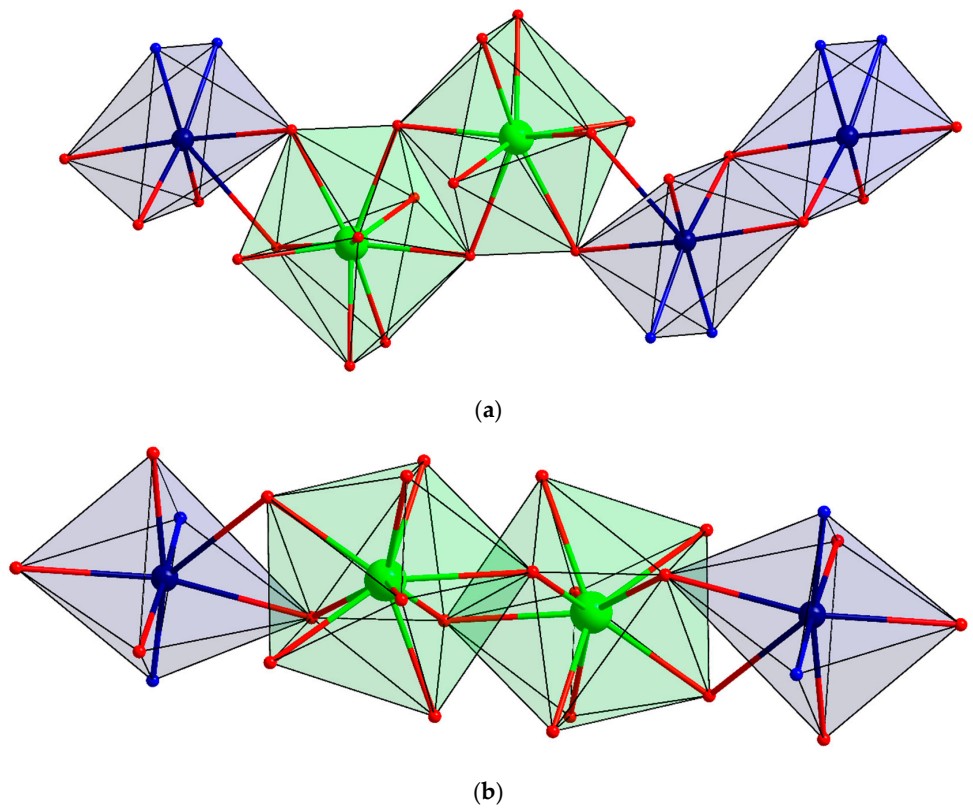

(a)

(b)

**Figure 2.** Coordination polyhedra of metal ions in I (**a**) and II (**b**).

**Table 1.** Selected bond lengths, the shortest interatomic distances $d$ (Å) and angles $\omega$ (°) in structures I and II.

| Compound/Parameter | Complex I | Complex II |
|---|---|---|
| Bond | | $d$ |
| Cd-O (O$_2$CR) | 2.254(4)–2.506(3) | 2.259(6)–2.778(7) |
| Cd-N (Phen) | 2.286(4), 2.313(4) | 2.321(7), 2.345(7) |
| Ln-O (O$_2$CR) | 2.364(3)–2.576(3) | 2.328(6)–2.437(6) |
| Ln-O (NO$_3$) | 2.448(4), 2.487(4) | - |
| Ln-N (MeCN) | - | 2.566(7) |
| Cd . . . Eu | 4.093(1) | 3.963(1) |
| Cd . . . Cd$_{min}$ | 3.820(1) | 7.237(1) |
| Ln . . . Ln | 4.042(1) | 4.469(1) |
| Angles | | $\omega$ |
| Cd-Eu-Eu | 119.21(2) | 163.87(2) |
| Cd-Cd-Ln | 112.01(2) | - |

A detailed analysis of the {Eu$_2$Cd$_2$} metal core geometric parameters in I and II shows a decrease in the angle values in tetranuclear fragments (from ~163° in II to 119° in I), a decrease in the Eu . . . Eu distance (from 4.469(1) Å in **II** to 4.042(1) Å in **I**), and an increase in the Cd . . . Eu distance (from 3.963(1) Å in II to 4.093(1) Å in I).

The geometry of the metal framework is affected by intramolecular secondary interactions associated with dinitrobenzoate anions (Tables S3–S6). 2,4-Dinitrobenzoate anions in **I** are involved in a significant number of intramolecular interactions: N-O . . . $\pi$ (Table S3), C-H . . . O (Table S4), NO$_2$ . . . .NO$_2$ (Table S5), and $\pi$-$\pi$ (Table S6). The geometric parameters and the large number of these interactions allow us to assume that their strength and contribution to the formation of the polymer structures synthesized are considerable. The solvate molecule of acetonitrile is imbedded in the crystal lattice of I by C-H...N and N-O...$\pi$-interactions. A different scheme of non-covalent interactions is observed in the case of the 3,5-dinitrobenzoate anion (Tables S3–S6): the aromatic anion is not involved in $\pi$-$\pi$-interactions with the phenanthroline molecule. Apparently, a significant number of intramolecular interactions in I leads to the convergence of aromatic fragments and zigzag arrangement of metal atoms that makes a significant contribution to the stabilization of the polymer structure (Figure S3). A similar distortion of the metal core and close values of interatomic bond lengths (Cd-O, Ln-O, Cd-N) was observed earlier in polymeric pentafluorobenzoate complexes with aromatic N-donor ligands where the polymer chain was stabilized by C-H . . . F, C-F . . . $\pi$, F-F, and $\pi$-$\pi$-interactions [9–11]. In the case of complex II, the fragments of 3,5-Nbz-anions are located fairly freely. The position of Phen molecules is almost perpendicular to the planes of the aromatic rings in the carboxylate anions, while intramolecular $\pi$-$\pi$-interactions are observed only between the N-donors' $\pi$-systems (Figure S4). A similar situation was observed in molecular pentafluorobenzoate complexes obtained at elevated temperatures [9].

It is interesting to analyze the geometry of the dinitrobenzoate fragments. Obviously, 3,5-dinitrobenzoate is flat [35,36], but the appearance of a nitro group at an ortho position leads to a deviation of the corresponding nitro group and carboxylate group from the planar arrangement. The torsion angles characterizing the positions of nitro and carboxy groups relative to the benzene ring plane are given in Table 2. Quantum-chemical calculations (pbe0, def2tzvp) were carried out in order to search for the most energetically favorable conformation of the corresponding fragments. It was taken into account that a change in the rotation angle of the carboxy group provokes a change in the rotation angle of the o-nitro group. The data obtained is shown in Figure 3. The upper graph shows the dependence of the energy on the OCCC angle, while the lower graph shows the changes in the ONCC angle. According to the data obtained, the most favorable conformation of 2,4- dinitrobenzoate is achieved if the OCCC angle is 120° and the ONCC angle is 31.9° (in our complex, it is implemented in the fragment O1–C7 and involved in N(3)–O(3) . . . $\pi$(2,4-Nbz) and two CH . . . O (C(4)–H(4) . . . O(4) C(14)–H(14) . . . O(3)) interactions). The conformation with the maximum energy is realized in the fragment O16–C21, the OCCC angle is 169.2° and the ONCC angle is −81°. Oxygen atoms of the nitro-group in this case participate in NO$_2$ . . . NO$_2$ and Lp . . . $\pi$ interactions. Both of these interactions stabilize the almost perpendicular position of the nitro group relative to the benzene cycle. Similar calculations for 3,5-dinitrobenzoate confirmed that the planar conformation was most favorable.

According to IR-spectral data, absorption bands related to stretching ($v$, st) symmetric (sy), asymmetric (as), and deformation ($\delta$) vibrations can be found in the spectra [37]. The bands at 3095–3098 cm$^{-1}$ (C–Hst) correspond to the vibrations of bonds in the aromatic rings. The vibrations of COO and NO$_2$ groups of benzoate anions manifest themselves as bands at 1612–1616 cm$^{-1}$ (C = Ost), 1580–1594 cm$^{-1}$ (N = Ost; $v_{as}$(COO$^-$)), 1536–1539 cm$^{-1}$ ($v_{as}$(NO$_2$)), 1461–1466 cm$^{-1}$ ($v_{sy}$(COO$^-$)), and 1341–1345 cm$^{-1}$ ($v_{sy}$ in NO$_2$). The vibrations of acetonitrile functional groups manifest themselves as a band at 1403 cm$^{-1}$ ($\delta_{sy}$ in CH$_3$) for I. The other signals correspond to MeCN, and the aromatic fragments can overlap with $v_{sy}$(COO$^-$) and $v_{as}$(COO$^-$).

**Table 2.** Selected torsion angles in compound I.

| O1-C1-C2-C3 | −122.3(5) | O16-C15-C16-C17 | 169.2(5) |
|---|---|---|---|
| O2-C1-C2-C3 | 60.8(6) | O17-C15-C16-C17 | −11.2(8) |
| C2-C3-N3-O4 | −155.3(5) | O19-N8-C17-C16 | −81.0(6) |
| C2-C3-N3-O3 | 25.4(7) | O18-N8-C17-C16 | 103.8(6) |
| O7-C8-C9-C10 | −36.4(7) | O23-C22-C23-C24 | −106.1(7) |
| O8-C8-C9-C10 | 148.7(5) | O22-C22-C23-C24 | 79.6(7) |
| O9-N5-C10-C9 | −41.9(6) | O24-N10-C24-C23 | 16.7(8) |
| O10-N5-C10-C9 | 141.3(5) | O25-N10-C24-C23 | −163.4(5) |

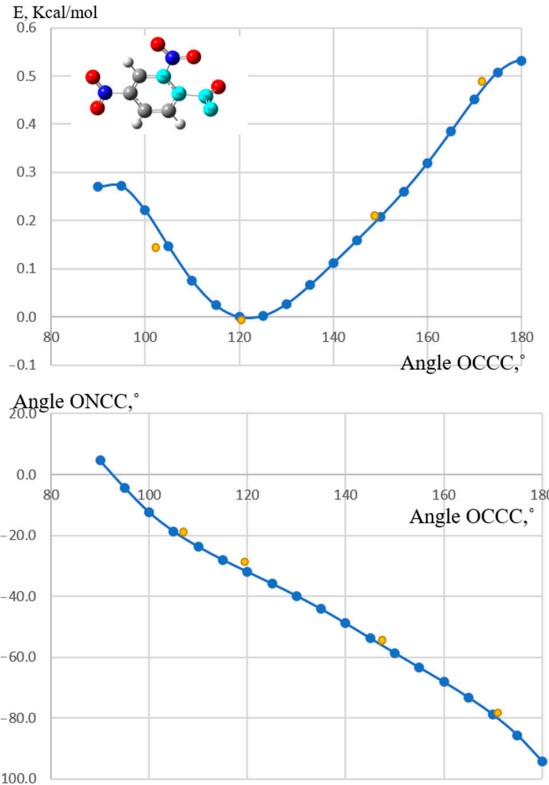

**Figure 3.** Quantum-chemical analysis of the dependence of energy on the angle of rotation of the carboxyl group in 2,4-dinitrobenzoate. **Above**—energy dependence, **below**—ONCC angle dependence.

The luminescent properties were studied for the compounds I and II obtained. However, emission of the europium ion was not observed at a wavelength of 270 nm, which is due to excitation energy dissipation, and is consistent with the previously obtained data for dinitrobenzoate complexes of 4f-ions [23–25].

### 3.2. Thermal Decomposition

DSC, TGA and DTA experiments at a linear heating rate of 5 °C/min were carried out to study the thermal decomposition processes in the title complexes I and II.

TGA and DTA analyses was performed for $[Eu_2(NO_3)_2Cd_2(Phen)_2(2,4\text{-Nbz})_8]_n$ ·$2n$MeCN (I). Figure 4 shows a small weight loss process that starts at 116 °C, ends at 165 °C, and is accompanied by 2.9% weight loss. It is attributed to the loss of solvate acetonitrile molecules (Calc. 2.9%). The main decomposition occurs at 267 °C with 57% weight loss. The DTA curve shows an exothermic process with a max peak temperature of 328 °C (Figure 5). The difference in Figures 4 and 5 is due to registration with a sharp

increase in temperature as a result of the exothermic effect. In this case, Figure 5 will be more indicative, which shows the dependence of the mass on time, and not on temperature, which makes it possible to more correctly show the change in the mass of the sample.

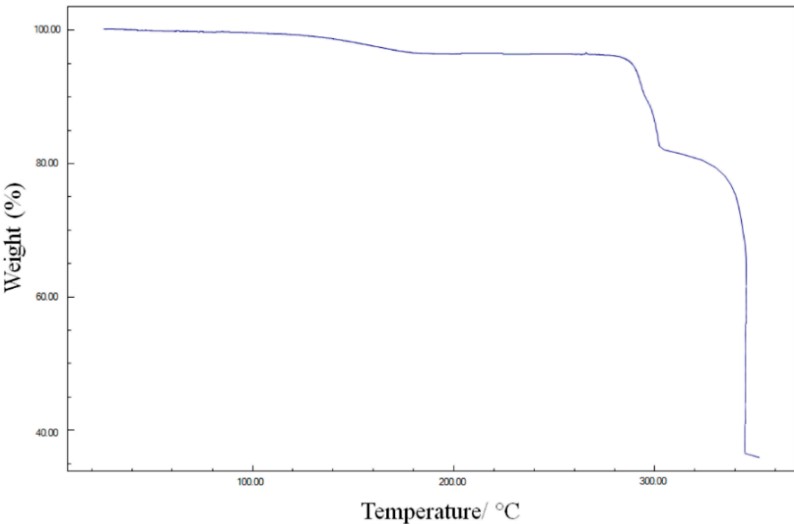

**Figure 4.** TGA curve of I.

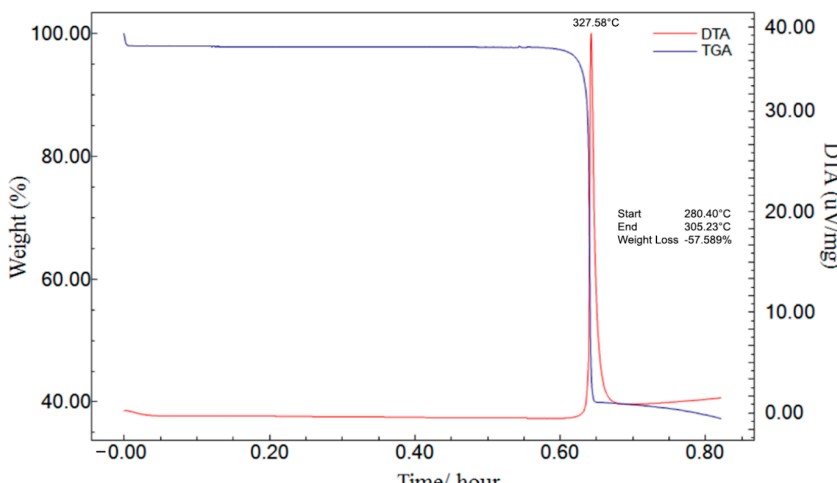

**Figure 5.** TGA and DTA curves of I.

TGA, DTA and DSC analyses were performed for [Eu$_2$(MeCN)$_2$Cd$_2$(Phen)$_2$(3,5-Nbz)$_{10}$] (II) (Figures 6, 7 and S5).

Figure 6 demonstrates the TGA curve of compound II. It shows a weight loss process starting at 73 °C, ending at 103 °C, and accompanied by 2.9% weight loss, which is attributed to the loss of coordinated acetonitrile molecules (Calc. 2.66%). The second weight loss process starts at 286 °C and is followed by the explosion of the sample of the compound.

As shown in Figure 7, compound II shows an endothermic broad peak around 97.91 °C corresponding to the a loss of coordinated acetonitrile molecules, which starts at 84 °C and ends at 110 °C. An intense exothermic process is observed from 295 °C to 310 °C, with the peak temperature of 303.92 °C, which indicates the main decomposition reaction with explosion of the sample.

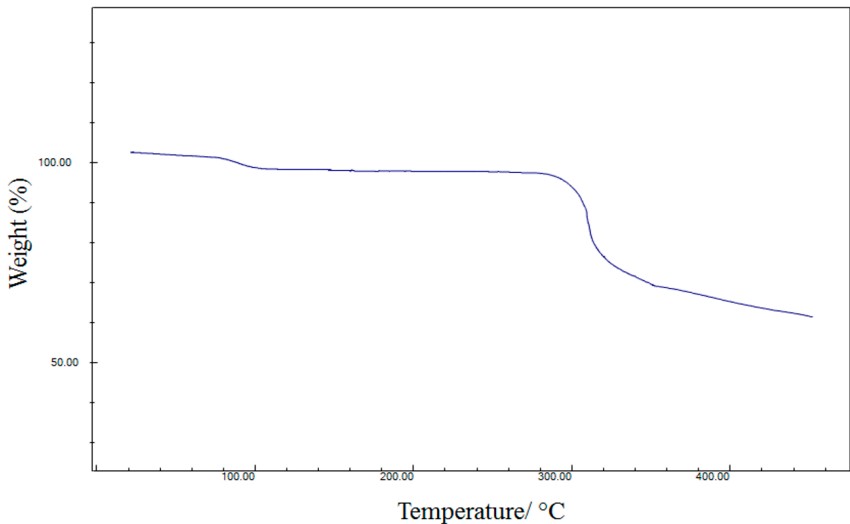

**Figure 6.** TGA curve of II.

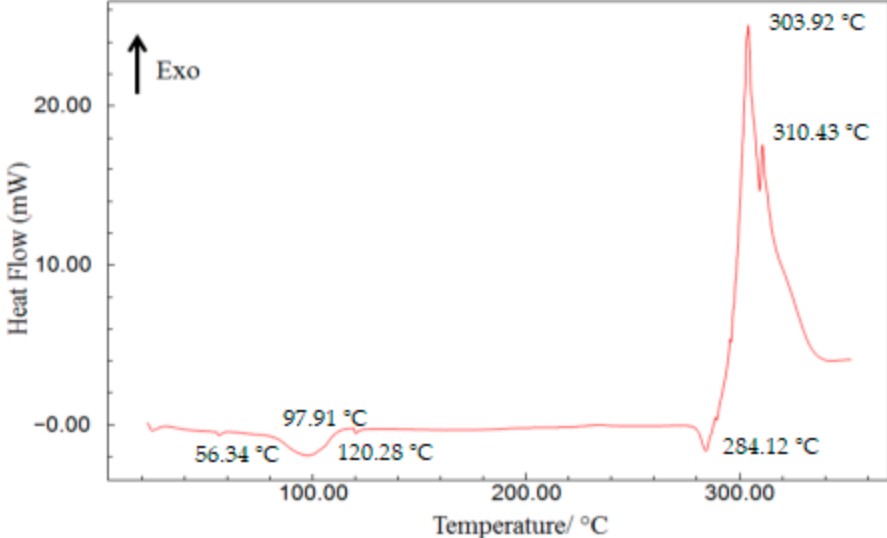

**Figure 7.** DSC curve of II.

It appears that, the different thermal behavior of the obtained compounds occurs due to the different scheme of non-covalent interactions in the crystal packings of complexes I and II. In the case of compound I, a denser crystal packing is formed, stabilized by a number of non-covalent interactions, and the solvate acetonitrile molecule is involved in N–O . . . π, C–H . . . O interactions, which leads to an increase in the loss temperature of the acetonitrile molecule compared to compound II.

Similar thermal effects are observed for a number of lanthanide compounds with 2,4- or 3,5-dinitrobenzoic acid [23], while explosion was not observed for the transition metal complexes with a similar structure [34,38].

Similar compounds with different isomers of dinitrobenzoic acid are known; these compounds can decompose both without [34,39] and with explosion [12,23]. The thermal stability can be affected by various factors, such as: the coordination number and coordination environment [40], the number of bridging ligands and the nature of lanthanide ions [23], numerous inter- and intramolecular hydrogen bonds [34,40], and architectures of the structure [13,23]. Both compounds are 1D polymers, so they explode at relatively low temperatures since lanthanide dinitrobenzoate compounds have the following trend: 3D polymer complexes are more stable than 2D ones, while 1D structures become more stable than 2D after the loss of the solvent [23].

## 4. Discussion

The demonstrated trend of 2,4-dinitrobenzoates to form polymeric structures is also typical of other complexes of metal atoms that can build donor-acceptor bonds of sufficient length [40–42]. Previously, using 3,5- and 2,4-dinitrobenzoate complexes of lanthanides with 1,10-phenatroline as examples, it was shown that the position of the substituent in the aromatic ring strongly affects the structure of the resulting complexes: 3,5-dinitrobenzoates are binuclear complexes, $[Ln_2(Phen)_2(3,5-Nbz)_6]$ (Ln(III) = Dy, Tb, Gd, Eu, Sm, Ce) [43–48], whilst 2,4-dinitrobenzoates with the same composition $[Ln_2(Phen)_2(2,4-Nbz)_6]_n$ have a polymeric structure [41]. Moreover, a polymeric complex is formed in the case of the cadmium compound, $[Cd(Phen)(2,4-Nbz)_2]_n$ [40]. It was found that stabilization of the polymer structure is associated both with intramolecular $\pi \ldots \pi$-interactions between the aromatic fragments of Phen and 2,4-Nbz, along with N-O $\ldots \pi$-interactions. On the other hand, the crystal packing in the structure of the molecular complex is stabilized due to $\pi \ldots \pi$-interactions between the aromatic fragments of 1,10-phenanthroline, while 3,5-dinitrobenzoate anions are not involved in $\pi \ldots \pi$-interactions.

A similar trend to form polymeric structures was also observed for pentafluorobenzoate complexes $[Cd(Phen)(C_6F_5COO)_2]_n$ and $[Eu_2Cd_2(Phen)_2(C_6F_5COO)_{10}]_n \cdot 3n$MeCN, where non-covalent interactions affected the geometry of metal fragments and led to stabilization of the polymer structures [9,11].

However, in such complex systems as compounds I and II, all noncovalent intramolecular/intermolecular interactions and the effects of electron density redistribution must be considered. Thus far, only the difference in the results of combinations of 1,10-phenanthroline with dinitrobenzoate anion isomers can be stated.

## 5. Conclusions

A detailed analysis of non-covalent interactions shows that the packing and structure of the complex can be controlled by changing the position of the nitro group in the benzoate anion. It has been demonstrated for the complexes obtained that, although the nitro group is not involved in coordination with metal atoms, it still contributes to the stabilization of the complex crystal packing due to non-covalent interactions and affects the structure of the compounds formed. Non-covalent interactions lead to a significant distortion of the metal core geometry and stabilize the polymeric structure in the case of the 2,4-dinitrobenzoate complex. Similar effects were observed with pentafluorobezoate analogs. In the case of the 3,5-dinitrobenzoate complex, a molecular structure that a typical of compounds with a similar composition is formed. It was shown that no metal-centered luminescence at a wavelength of 270 nm, which is associated with the dissipation of the excitation energy, is observed in the resulting complexes. A study of the thermal stability shows that the compounds lose solvent molecules and then remain stable up to 260 °C and 295 °C for compounds I and II, respectively, after which exothermic decomposition occurs.

**Supplementary Materials:** The following are available online at https://www.mdpi.com/article/10.3390/cryst12040508/s1, Figures S1 and S2 (PXRD data); Tables S1–S6, Figures S3 and S4 (structural data); Figure S5 (Differential thermal analysis (DTA) data); Figure S5: Differential thermal analysis (DTA) data for compound II.

**Author Contributions:** Conceptualization, validation, M.A.K., A.A.S., N.V.G.; methodology, M.A.S., N.V.G., A.A.S.; formal analysis, M.A.S., A.S.C., J.K.V., F.M.D.; investigation, M.A.S., G.A.R., V.V.K.; writing—original draft preparation, M.A.S., A.A.S.; writing—review and editing, M.A.K.; supervision, I.L.E. All authors have read and agreed to the published version of the manuscript.

**Funding:** This research was funded for Grants of the President of the Russian Federation, grant number MK-94.2022.1.3.

**Data Availability Statement:** The data presented in this study are available in this article and Supplementary Materials.

**Acknowledgments:** X-ray diffraction analysis, CHN and IR-spectral analyzes were performed using the equipment at the Center for Collective Use of the Kurnakov Institute RAS, which operates with the support of the state assignment of the IGIC RAS in the field of fundamental scientific research.

**Conflicts of Interest:** The authors declare no conflict of interest.

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
