# Peer review of "Effect of Non-Covalent Interactions on the 2,4- and 3,5-Dinitrobenzoate Eu-Cd Complex Structures"

_crystals, doi:10.3390/cryst12040508_

Round 1
Reviewer 1 Report
The presented work discuss non-covalent interactions effect on the structure of Eu-Cd complex of dinitrobenzoates. The introduction is very short and can be extended. The synthesis part is well written and does not require any changes. Discussion is also well presented and different aspects are taken into account.
There are small editorial mistakes that requires to be changes. For example:
- Fig 2 in description I and II are not written in bold
- Fig 3 – decimals needs to be corrected (points instead of comas)
Author Response
Q.: Fig 2 in description I and II are not written in bold
- Fig 3 – decimals needs to be corrected (points instead of comas)
Answer: The Manuscript was corrected.
Reviewer 2 Report
In the above-mentioned manuscript, the authors report syntheses, single crystal X-ray diffraction structure analyses, IR spectroscopy, and the thermal stability for two heterometallic complexes. The basic characterization experiments have been well done, the crystal structure analysis is clear, the discussion is reasonable and the conclusion is supported by enough evidences. The work is able to attract enough attention from the colleagues in the community, including inorganic and coordination chemistry, accordingly, I recommend acceptance of this paper for publication after minor revision.
(1) IR spectra of complexes have been widely investigated so far, and some characteristic IR vibrational bands have been clearly assigned, however, the authors have not cited any references for the assignments of the bands in IR spectra, thus, the suitable references should be provided for IR vibrational bands assignment.
(2) The author emphasizes the thermal stability of the obtained com-pounds, but what are the powder residues? Perhaps the authors collected the powder pattern, the confirmation data (PXRD) should be included the supporting information.
(3) There are some noticeable formatting and grammatical errors in the manuscript. For example, Page 3 line 143 “three chelate-bridged 2,4-NO2 anions, and one chelated nitrate-anion ()”. Abstract “[Eu2(NO3)2Cd2(Phen)2(2,4-Nbz)8]n.2nMeCN (I) and [Eu2(MeCN)2Cd2(Phen)2(3,5-Nbz)10]” Check the format and all English carefully.
Author Response
Q.: (1) IR spectra of complexes have been widely investigated so far, and some characteristic IR vibrational bands have been clearly assigned, however, the authors have not cited any references for the assignments of the bands in IR spectra, thus, the suitable references should be provided for IR vibrational bands assignment.
Answer: The reference was added.
Q.: (2) The author emphasizes the thermal stability of the obtained compounds, but what are the powder residues? Perhaps the authors collected the powder pattern, the confirmation data (PXRD) should be included the supporting information.
Answer: Unfortunately, we were not able to analyze the product remaining after thermal analysis because we used small amounts of samples (6 mg). Large part of the samples was lost due to the “rupture” of the aluminum bowl after the observed large exothermic effect.
Q.: (3) There are some noticeable formatting and grammatical errors in the manuscript. For example, Page 3 line 143 “three chelate-bridged 2,4-NO2 anions, and one chelated nitrate-anion ()”. Abstract “[Eu2(NO3)2Cd2(Phen)2(2,4-Nbz)8]n.2nMeCN (I) and [Eu2(MeCN)2Cd2(Phen)2(3,5-Nbz)10]” Check the format and all English carefully.
Answer: The Manuscript was corrected.
Reviewer 3 Report
Brief Summary
The manuscript, Effect of non-covalent interactions on the 2,4- and 3,5-dinitrobenzoate Eu-Cd complex structures, describe the effect of manipulating the structure and its possible impact on its properties, particularly thermal stability in this case. While the research work is impressive, it is unclear why Eu or 2,4-dinitrobenzoic and 3,5-dinitrobenzoic acids were selected in this study. In general, there are many minor grammatical errors that need to be corrected before publication.
line 11-13
Comment: The abstract, results, and discussions of the different analyses (IR, elemental, DSC/ DTA/ TGA, PXRD, single crystal) listed in the paper should be modified to a particular sequence to benefit the reader.
line 27-28: “promising for solving various practical problems”.
Comment: A few examples/ citations will be beneficial in the introduction
line 39-43: “While investigating fluorobenzoate complexes, it is excluded the capability to trace the influence of position halogen substituents, since the structure-forming effects of non-covalent interactions characteristic of pentafluorobenzoate systems do not appear when passing to benzoic acidic substituents with a smaller number of fluorine atoms [7]. Anions of 2,4-dinitrobenzoic (2,4-Nbz) and 3,5-dinitrobenzoic (3,5-Nbz) acid are promising objects for solving this problem.”
Comment: These lines are particularly confusing since 2,4-dinitrobenzoic (2,4-Nbz) and 3,5-dinitrobenzoic (3,5-Nbz) acids will not have any halogen bonding non-covalent interactions. If the focus is to emphasize the effect of the position it needs to be worded differently, since it’s unclear how 2,4-dinitrobenzoic (2,4-Nbz) and 3,5-dinitrobenzoic (3,5-Nbz) acid is solving this problem.
Comment: While not important, I suggest including a paragraph in the introduction on why Eu is a good candidate for this study.
line 64; Comment: mentions Phenanthroline monohydrate while in line 122 its 1,10-Phenanthroline.
line 71-73; Comment: Only DSC and DTA are listed here, while there is TGA data in the paper as well.
line 136; Comment: Was any other solvent other than Acetonitrile used in this study? It might be useful to see the effect of solvent inclusion if any in the structure-property relationships.
line 194 “was stabilized by C-H…F, C-F… π, F-F and π-π interactions [5-7]”.
Comment: Are the bond lengths comparable?
line 200-201 “Obviously, 3,5-dinitrobenzoate is flat”
Comment: Might be obvious, but citations are important.
line 226-227 “was performed only DTA analysis”.
Why was DSC not performed? line 226 mentions that only DTA was performed while TGA results are also included.
line 230 “Figure 5, with max peak temperature at 328°C”
Comments:
- No temperature axis is included in the figure, unsure how to read 328 C.
- Why do TGA curves in Figures 4 and 5 visually have significant weight loss differences?
- Figures 4, and 5 should include peak labels for the starting and ending Temperature points and also weight loss. The same is true for Figures 6 and 7.
- TG and TGA are alternatively used, please keep it consistent throughout. I prefer the latter
- Why was DTA not performed for compound II?
line 227 and 238:
Comment: For compound, I ACN loss starts at 116°C, ends at 165°C, for compound II ACN loss starts at 73 C, ends at 103 C. Please explain why this difference was observed.
lines 244-248: “can be influenced by various factors, such as coordination numbers and co-ordination environment [27], the number of bridging ligands and the nature of the lanthanide ions [13], extensive inter- and intra-molecular hydrogen bonds [24, 27], architectures of the structure [8, 13]”
Comments: The remark on higher stability is discussed in general, it is important that the authors point out why the difference was observed in their case given the information they have. The thermal stability of compound I could be possibly explained due to comparatively smaller bond lengths and higher intramolecular interactions. I suggest including a comment on this
line 255-256 “can build donor-acceptor bonds of sufficient length”.
Comment: Citations required
line 105 “Anal. Calc.”
Comment: please write in complete or make a list of abbreviations.

Author Response
Thank you very much for your reviewing. Enclosed please find the reply to your comments.

Round 2
Reviewer 1 Report
After the authors have changed and supplemented the work, it may be published.
Reviewer 3 Report
Thank you for your detailed answers to the comments and for adding the required changes.